# Effects of Orange Peel Extract on Laccase Activity and Gene Expression in *Trametes versicolor*

**DOI:** 10.3390/jof10060370

**Published:** 2024-05-22

**Authors:** Simon Vandelook, Berend Bassleer, Elise Elsacker, Eveline Peeters

**Affiliations:** Research Group of Microbiology, Department of Bioengineering Sciences, Vrije Universiteit Brussel, Pleinlaan 2, B-1050 Brussels, Belgium; simon.vandelook@vub.be (S.V.); elise.vanden.elsacker@vub.be (E.E.)

**Keywords:** laccase, *Trametes versicolor*, white-rot fungi, orange peels, transcriptional expression

## Abstract

The genome of *Trametes versicolor* encodes multiple laccase isozymes, the expression of which is responsive to various conditions. Here, we set out to investigate the potential of orange peel extract as an inducer of laccase production in this white-rot fungus, in comparison to the previously identified inducing chemical compound, veratryl alcohol. For four geographically distinct *T. versicolor* strains, a positive correlation has been observed between their oxidative activity and incubation time in liquid cultures. The addition of 20% orange peel extract or 5 mM veratryl alcohol caused a rapid increase in the oxidative potential of *T. versicolor* M99 after 24 h, with a more pronounced effect observed for the orange peel extract. To elucidate the underlying molecular mechanisms of the induced laccase activity, a transcriptional gene expression analysis was performed for the seven individual laccase genes in *T. versicolor*, revealing the upregulation of several laccase genes in response to the addition of each inducer. Notably, the gene encoding TvLac5 demonstrated a substantial upregulation in response to the addition of 20% orange peel extract, likely contributing to the observed increase in its oxidative potential. In conclusion, our results demonstrate that orange peels are a promising agro-industrial side stream for implementation as inducing agents in large-scale laccase production with *T. versicolor*.

## 1. Introduction

*Trametes versicolor* is a wood-degrading fungus belonging to the Polyporaceae. It is classified as one of the white-rot fungi (WRF), which are excellent wood decomposers due to their ability to degrade cellulose, hemi-cellulose and lignin components in plant cell walls [1]. This is mediated by the secretion of powerful hydrolytic and oxidative enzymes into the surrounding environment [2]. The ligninolytic systems of WRF comprise several key extracellular enzymes, namely dye-decolorizing peroxidases (EC 1.11.1.19, DyP), lignin peroxidases (EC 1.11.1.14, LiP), manganese peroxidases (EC 1.11.1.13, MnP), versatile peroxidases (EC 1.11.1.16, VP) and laccases (EC 1.10.3.2, Lac) [3]. Laccases form a family of blue multi-copper oxidases, named after the intense blue color associated with the notable absorption spectrum arising from their multi-copper core, which underlies their oxidative potential [4]. These enzymes exhibit a large commercial potential, given that laccase-catalyzed degradation reactions do not require the addition of hydrogen peroxide [5]. Besides lignin, laccases have the potential of oxidizing a wide range of recalcitrant compounds, such as aromatic hydrocarbons, including phenols, polyphenols, aminophenols and benzenethiols; they can also oxidize polyamines, hydroxyindoles, anilines and numerous inorganic ions. They function by reducing oxygen as a co-factor and creating water as a by-product [6,7,8].

Although commonly found in WRF, laccases have also been identified in other phylogenetically diverse eukaryotic lineages and even in a subset of prokaryotes [9]. Different families are involved in diverse biological functions, participating in either catabolic or anabolic pathways [6]. In plants, laccases were found to be involved in tissue lignification by catalyzing the polymerization of natural phenolics [10]. On the contrary, fungal laccases participate in delignification reactions during wood degradation [11,12,13,14]. These differences in enzymatic activity are underscored by markedly different redox potentials. While bacterial and plant laccases have a low potential, fungal laccases are typically characterized by a medium-to-high redox potential for non-wood- and wood-degrading fungi, respectively [6]. A high oxidation potential is directly linked to the high efficiency and substrate diversity of laccase enzymes; hence, laccases from wood-degrading fungi, including *T. versicolor*, are highly desired for biotechnological applications [15]. Laccases are finding various applications, such as their use as cross-linking agents in furniture fabrication, as food additives in the processing of beverages (e.g., wine, beer, orange juice and tea), for processing steps in pulp, paper and the textile industry, as well as in applications in nanobiotechnology (e.g. biosensor development), synthetic chemistry, cosmetics and the bioremediation of soil and water [16,17]. Regarding the latter, it is of interest that laccases degrade polycyclic aromatic hydrocarbons (PAHs) and other recalcitrant organic pollutants [18,19].

WRF typically encode multiple laccase isozymes in their genomes. These laccase genes are not genetically redundant but are characterized by distinct expression and regulatory profiles in response to environmental signals. Although some laccases partake in lignin degradation, others can be involved in diverse developmental stages and biological roles such as defense, sporulation, pigmentation or pathogenesis [20]. It has been reported that, upon growth in submerged cultures, laccase expression is regulated in response to the nitrogen and carbon sources used, as well as to the carbon-to-nitrogen ratio [21,22,23,24]. In addition, laccase production can be induced upon an increased Cu^2+^ concentration [23,24,25] and the presence of different aromatic molecules in the medium [25,26,27,28]. This can be explained by Cu^2+^ being a direct constituent of the laccase enzyme. Indeed, depending on the laccase type, 3 or 4 Cu^2+^ ions are integrated into the active site [25].

With seven laccase-encoding genes, *T. versicolor* has been recognized as one of the most effective WRF for large-scale laccase production [29,30,31]. From a biotechnological perspective, it is of great interest to identify inducers for laccase production, whether these are chemical inducers, such as aromatic compounds, alcohols, detergents or metal ions; or process conditions such as aeration [32,33,34]. Thus far, regarding *T. versicolor*, gene expression studies have been mostly focused on analyzing the inducing effect of chemicals such as veratryl alcohol, benzo(a)pyrene or copper-glycyl-L-histidyl-L-lysine for improved laccase production, each causing a distinct expression pattern [18,35]. Based on these results, it has been hypothesized that three laccases, TvLac2, TvLac3 and TvLac6, perform an extracellular function, such as lignin degradation, while three others, TvLac1, TvLac4 and TcLac7, function intracellularly in lignin catabolism. Finally, the TvLac5 isoform was previously found to be strongly transcriptionally induced in response to the presence of the PAH compound benzo(a)pyrene [18].

Besides chemical compounds, it is of great interest to identify the agro-industrial side streams or waste streams as natural inducing agents for laccase production in *T. versicolor*. It was previously reported that grounded orange peels induce laccase production in the related species *Trametes hirsute* [36]. In this study, we set out to investigate the effect of extracts of orange peels, a natural source containing a complex composition of diverse aromatic compounds and nutrients, on the extracellular laccase activity of *T. versicolor*. Focusing on the *T. versicolor* strain M99, we have set out to quantitively determine the transcriptional regulatory effects in response to the presence of orange peel extract (OPE) on each of the seven laccase genes.

## 2. Materials and Methods

### 2.1. Strains and Culturing Conditions

Four *T. versicolor* strains were used in this study. Three strains were purchased from depositories, whereas the remaining strain was isolated from nature (Table 1). The wild isolate was determined to be a *T. versicolor* strain by Internally Transcribed Spacer (ITS) sequencing, followed by a BLAST search against the NCBI database.

All strains were maintained on a solid medium with 1% malt and 1% agar (malt agar medium) at 4 °C. Cultivation on the solid medium was performed on malt agar medium with 1–2% malt, at 26 °C, in the dark. When indicated, one of the following supplements were added: 20% OPE or 5 mM veratryl alcohol (Sigma-Aldrich, Burlington, MA, USA). OPE was prepared by autoclaving 30 g wet orange peels in 1 L double-distilled water (ddH_2_O) for 15 min. Plates were inoculated by adding 100 µL of a homogenous hyphal cell suspension, previously obtained by mechanical disruption of hyphal pellets with a glass tube tissue homogenizer.

Liquid cultures were prepared in Trametes Defined Medium (TDM) [37] containing 10 g/L glucose, 2.1 g/L NH_4_Cl, 10 mg/L thiamine, 100 mL/L macronutrients (20 g/L KH_2_PO_4_, 5 g/L MgSO_4_·7H_2_O, 1 g/L CaCl_2_) and 10 mL/L micronutrients (1.5 g/L nitrilotriacetic acid, 3 g/L MgSO_4_·7H_2_O, 0.5 g/L MnSO_4_·H_2_O, 1 g/L NaCl, 0.1 g/L FeSO_4_·7H_2_O, 0.1 g/L CoSO_4_·7H_2_O, 0.1 g/L CaCl_2_·2H_2_O, 0.01 g/L CuSO_4_·5H_2_O, 0.01 g/L AlK(SO_4_)_2_·12H_2_O, 0.01 g/L H_3_BO_3_, 0.01 g/L NaMoO_4_·2H_2_O). The pH was adjusted to 4.5 with 0.5 M NaOH. If indicated, 20% OPE or 5 mM veratryl alcohol (Sigma-Aldrich, Burlington, MA, USA) was added to the medium. Subsequently, the medium was autoclaved for 15 min at 121 °C, and 25 µg/mL streptomycin was added to prevent potential bacterial contamination. The inoculum was prepared by blending a fully grown 1% malt extract agar plate with 100 mL sterile ddH_2_O in a stainless steel semi-micro blending container of 360 mL (Eberbach corporation, Van Buren Charter Township, MI, USA) for 10 s. Afterwards, 5 mL of this inoculum was added to the liquid medium, and cultures were incubated in a shaking incubator at 130 rpm, at 26 °C and in the dark.

For all liquid cultures, biomass quantification was performed by performing dry weight measurements. To this end, mycelium pellets were separated from the liquid medium by performing vacuum filtration of the cultures using Whatman filter paper (GE healthcare, Buckinghamshire, UK). The biomass was rinsed off from the filter paper using ddH_2_O, and collected. Samples were dried in a ventilated oven at 60 °C for 24 h.

### 2.2. Enzymatic Assays

To determine oxidative activity, culture samples were subjected to colorimetric enzymatic assays performed with the redox substrate 2,2′-azino-di(3-ethylbenzothiazoline-6-sulfonic) (ABTS) (Sigma-Aldrich, Burlington, MA, USA). ABTS oxidation is specific for laccase activity in the absence of H_2_O_2_ [38]. An ABTS reaction mixture was prepared, consisting of 14 mM ABTS in 50 mM glycine-HCl at pH 3 [39]. Oxidation profiles were determined by measuring the absorbance at a wavelength of 436 nm (A_436_). In a first experimental set-up, 0.5 mL of culture filtrate was combined with 0.5 mL of ABTS reaction mixture (14 µmol ABTS), and absorbance was measured after a 5 min reaction time using a UV-1601PC spectrophotometer (Shimadzu, Kyoto, Japan). In a second set-up, a 1:1 culture filtrate and ABTS reaction mixture ratio was used in a total reaction volume of 0.2 mL (2.8 µmol ABTS), and measurements were performed in a 96-well plate in the Synergy H1 microplate reader (Biotek, Winooski, VT, USA). This allowed a time-course measurement of the reaction by measuring absorbance every 15 s during a total period of 5 min.

### 2.3. Quantification of Colonization

For quantification of colonization on solid medium, each plate was inoculated in the center with a single circular agar plug with a diameter of 4 mm taken from a fully colonized plate. Next, mycelial colonization was followed by taking a high-resolution image of the surface every 24 h using a flatbed scanner (CanoScan 9000F Mark II, Canon, Tokyo, Japan). Based on these images, the areas colonized by the mycelium were determined by adjusting contrast and brightness and by tracing the outline using the freeform tool in ImageJ2 [40]. The pixel-to-surface measurement tool in ImageJ2 was employed to convert this pixel tracing into quantitative data. Each condition was tested in triplicate. To determine statistical significance, a one-way ANOVA test was performed, with Tukey’s multiple comparison.

### 2.4. Nucleic Acid Extraction

After incubating the liquid cultures for 7 days, mycelial pellets were separated from liquid medium by collecting them on Whatman filter paper through vacuum filtration. Next, the mycelium was washed twice with ddH_2_O, and cell lysis was performed by grinding pellets in liquid nitrogen with a pestle and mortar until a fine powder was obtained.

DNA isolation was performed using a Wizard^®^ Genomic DNA Purification kit (Promega, Madison, WI, USA) according to the manufacturer’s instructions. RNA isolation was performed using a Nucleospin^TM^ RNA Plant and Fungi kit (Macherey-Nagel, Düren, Germany) according to the manufacturer’s instructions specific for mycelium tissue. Subsequently, RNA samples were treated with a TURBO™ DNase kit (Invitrogen, Waltham, MA, USA) to eliminate residual genomic DNA and were visualized using RNA gel electrophoresis to assess quality (Appendix A). RNA samples were stored immediately at −80 °C until further use.

### 2.5. Bioinformatic Analysis

To provide a comparative overview of the seven laccase proteins and their conserved domains in *T. versicolor*, laccase protein sequences of *T. versicolor* FP-101664 SS1 were retrieved from the NCBI databank, and a multiple sequence alignment was performed by MAFFT v7.427.

### 2.6. Quantitative Reverse Transcriptase PCR

Using total RNA as a template, cDNA was prepared with the iScript™ Reverse Transcription kit (Bio-Rad, Hercules, CA, USA) according to the manufacturer’s instructions. In the cDNA reaction mixtures, the RNA template concentration was normalized using the lowest RNA concentration as baseline. After synthesis, the obtained cDNA was diluted 10-fold in nuclease-free H_2_O and stored at −20 °C until further use.

For transcriptional gene expression analysis, quantitative Reverse Transcriptase PCR (qRT-PCR) was performed. The gene encoding β-tubulin 2 (NCBI ref AY944859) was selected as a housekeeping gene, enabling to measure relative expression levels by normalization. For this gene and each of the 7 laccase isozyme genes, primer pairs were designed to yield amplicons between 100 and 200 base pairs (bp) and to have a GC content between 50 and 60% (Table 2, Appendix A). This was achieved using the NCBI primer BLAST tool (https://www.ncbi.nlm.nih.gov/tools/primer-blast/, accessed 20 February 2019) [41]. The qRT-PCR reactions were prepared with a SYBR Green qPCR Master Mix (Promega) in 96-well plates as follows: 10 µL Master Mix, 0.8 µL forward primer (10 pmol/µL), 0.8 µL reverse primer (10 pmol/µL), 7.4 µL nuclease-free H_2_O and 1 µL cDNA dilution. Thermal cycling was performed in a MyiQ2 Real-Time PCR detection system device with the following parameters: initial denaturation at 95 °C for 3 min, followed by 40 cycles at 95 °C for 10 s and 55 °C for 30 s. Primer efficiencies were assessed through standard dilution series, followed by a melt curve analysis. Relative gene expression ratios were calculated according to the method of Livak and Schmittgen [42]. Each condition was represented by three biological replicates.

## 3. Results and Discussion

### 3.1. Growth and Oxidative Potential of Four Different T. versicolor Strains in Liquid Cultures

In an initial phase of this study, the extracellular oxidative potential was assessed for four different *T. versicolor* strains (Figure 1). The strains in this collection differ in geographic origin and in the method with which they were conserved (Table 1). While the strains M38 and M46 were conserved in propylene straws at −130 °C for several years, the strain M99, which is typically used in mushroom farming, was acquired as an actively cultured strain. Finally, the strain SV1, originally isolated from a dead tree trunk in a forest close to Brussels (Belgium), was also maintained as an active culture. SV1 was confirmed to be a *T. versicolor* strain by the direct sequencing of fungal ribosomal RNA genes [43].

A daily measurement was performed, starting on day 3 post inoculation and lasting until day 9 (Figure 1A). For all strains, a positive correlation was found between the total oxidative activity and the incubation time. For the later time points, this trend was less apparent for the strain M99, for which a decrease was noticed on days 8 and 9, and for the strain SV1, for which a plateau was reached on day 7. On day 9, the fungal biomass was recovered from the liquid cultures through vacuum filtration, and the amount of dry biomass was determined for each strain (Figure 1B). A similar amount of dry biomass was recovered for all strains, except for M46. Despite the reduced biomass produced by this strain, no negative effect was found on the accumulated oxidative potential over time, as compared to the other strains.

### 3.2. Impact of the Addition of Orange Peel Extract and Veratryl Alcohol on the Oxidative Potential

Subsequently, we investigated the effect of the addition of an OPE preparation on the oxidative potential of *T. versicolor* (Figure 2). This experiment was performed for the strain M99 of *T. versicolor*, which was selected because of its decreasing oxidative activity after an incubation of 7 days and longer (Figure 1A), thus representing an interesting candidate for further optimization by the induction of laccase production. As a reference, 5 mM veratryl alcohol was tested as well, which is a well-known inducer of laccase expression in WRF [26]. The inducers were added to 7-day old shaken cultures of the strain. In contrast to the previous set-up, the oxidative potential was measured over a shorter time frame, after 1 and 24 h after the addition of the inducer, and kinetic measurements were made (Figure 2).

One hour after the addition of the inducer, there was no noticeable change in the oxidative potential of the M99 strain for either OPE or veratryl alcohol (Figure 2A). In contrast, at 24 h after the addition, a clear change was observed in the oxidative potential (Figure 2B). The highest rate of enzymatic activity was measured for OPE, which caused a rapid increase of the oxidative potential, reaching saturation after 200 s. For veratryl alcohol, an induced oxidative potential was observed as well, although characterized by a lower enzymatic activity, as compared to OPE (Figure 2B).

### 3.3. Impact of Orange Peel Extract and Veratryl Alcohol on Surface Colonization and Biomass Yield

In other WRF species, veratryl alcohol, or its oxidized form, veratrylaldehyde, causes a growth-inhibiting effect at concentrations of 5 mM or higher [26,27]. To investigate this for *T. versicolor* and to compare this to OPE, we analyzed the effect of either 5 mM veratryl alcohol or 20% OPE on growth on a solid medium, by measuring the radial colonization rate (Figure 3). For the control condition, containing only TDM, all strains were able to fully colonize the surface of a petri dish within 6 days. Upon the addition of veratryl alcohol, a systematic decrease in the colonization rate was observed for all four strains (Figure 3A). In contrast, the addition of OPE caused only a slight decrease in the colonization rate for the strains M38 and M99, while not affecting the growth of the strains M46 and SV1. Statistical analysis confirmed a significant decrease (*p* < 0.01) in the colonization rate in the presence of 5 mM veratryl alcohol, as compared to the control condition (Figure 3B). In the presence of 5 mM veratryl alcohol, a significant decrease (*p* < 0.01) in the colonization rate was observed, as compared to the control condition (Figure 3B).

In the next step, the investigation was extended to liquid medium by cultivating *T. versicolor* M99 in TDM, either in the absence or presence of OPE, or with 5 mM veratryl alcohol (Figure 4). The addition of OPE colored the medium slightly orange, but this coloration faded away after 2 to 3 days of incubation. This decoloration was an indication of the activity of secreted enzymes [44,45]. After 7 days of cultivation, the cell dry weights were determined for the biomass produced in the different liquid cultures (Figure 4). This demonstrated a significant decrease (*p* ≤ 0.05) in biomass for the cultures in the presence of 5 mM veratryl alcohol, as compared to the cultures in the presence of OPE. Additionally, the average mycelium dry weight appeared higher for the control condition, as compared to the condition with veratryl alcohol, but without being supported by statistical significance.

The underlying mechanisms of the negative influence of veratryl alcohol on growth, on either solid or liquid media, could potentially be linked to an increased metabolic resource allocation for protein synthesis due to the induction of laccase production. In contrast, the lack of observing a similar negative effect for OPE might be attributed to an increased availability of nutrients for the organism (e.g., pectin, fructose, …), compensating for this negative effect. This growth-supporting role is enabled through the extraction of various reduced sugars and nutritive components during the hot water treatment of the OPE preparation. Orange peels are mainly composed of cellulose, which breaks down into glucose and fructose when hydrolyzed [46]. According to Yeoh et al., hot water treatment of orange peels enables yielding around 5% pectin on a dry basis, which is also a source of reduced sugars when hydrolyzed [47]. In conclusion, the addition of OPE did not significantly impact the radial colonization rate, while a negative impact was observed upon adding 5 mM veratryl alcohol.

### 3.4. Effects of Orange Peel Extract and Veratryl Alcohol on the Transcriptional Expression of Laccase Genes

Based on the increased oxidative potential caused by both OPE and veratryl alcohol (Figure 2), it can be hypothesized that the transcriptional expression of laccase genes is induced. The *T. versicolor* genome encodes seven different isozymes [29,30], of which genes encoding TvLac1, TvLac2 and TvLac3 are all spatially isolated from each other, while TvLac4 and TvLac5, and TvLac6 and TvLac7, respectively, are clustered in pairs (Figure 5). A multiple-sequence alignment enabled the identification of conserved Cu-binding residues in the active site for all isoforms (Appendix A). To correlate the increase in oxidative potential with a potentially higher relative transcription level of individual laccase genes in the presence versus absence of OPE or veratryl alcohol, a transcriptional gene expression analysis was performed by qRT-PCR (Figure 6).

The shaken submerged cultures of *T. versicolor* M99 were prepared with 20% OPE or 5 mM veratryl alcohol, added from the start of the experiment. The addition of OPE resulted in a significant transcriptional upregulation of the laccase genes *LCC2*, *LCC3* and *LCC5*, with log_2_ values of 3.03, 2.56 and 8.23, respectively (Figure 6). Similar observations were made for the addition of veratryl alcohol, albeit with lower log_2_-fold ratios (1.82, 3.33 and 2.65, respectively). Such similar differential gene expression profiles for OPE and veratryl alcohol suggest that aromatic compounds are largely responsible for this effect. Indeed, orange peels are rich in aromatic compounds, including free phenolic compounds from two main classes: flavonones and flavones. Moreover, large amounts of phytochemicals, such as carotenoids, flavonoids, aldehydes, esters, terpenes, alcohols and ketones, are also present in orange peels [48,49]. For example, limonene constitutes around 75% of the total extractable essential oils [46] and is therefore likely to be involved in influencing laccase expression, as previously observed by Böhmer et al. [50]. This can be explained by the structural resemblance between limonene and compounds found in lignin. Given the complex composition of OPE, it is difficult to quantify the precise involvement of each of its components in the more pronounced transcriptional induction.

*LCC2* and *LCC3* encode TvLac2 and TvLac3, respectively, which were previously shown to be secreted extracellularly, being involved in lignin degradation [18]. These laccase isoforms can be assumed to be partially responsible for the increased oxidative potential observed for both OPE and veratryl alcohol (Figure 2B). Ottini et al. also reported the upregulation of a laccase-encoding gene homologous to *LCC3* in *T. versicolor* MUM 04.100 when cultivated in the presence of alternative carbon sources [51].

In addition to *LCC2* and *LCC3*, the most pronounced upregulation in response to the presence of OPE was observed for *LCC5*, encoding TvLac5, with a log_2_ value of 8.23 corresponding to a fold-ratio value of 300. This is in agreement with the previous observation of benzo(a)pyrene strongly inducing the transcription of this isoform [18], thereby suggesting that *LCC5* transcriptional expression is highly responsive to xenobiotic compounds. Indeed, Yang et al. reported that binding sites for xenobiotic-responsive elements are present in the promotor region of the *lac5930-1* gene in *Trametes velutina*, and that these elements likely mediate a response to aromatic compounds [28]. The presence of xenobiotic-responsive elements was also predicted for the promoter sequence of *LCC5* in *T. versicolor* (Appendix A). Besides a TATA box, two putative metal-responsive elements (MREs) and eight putative CreA-binding sites were predicted, some of which might underly the observed transcriptional upregulation of *LCC5*. Given the large extent of this upregulation, as well as the significantly larger effect in response to OPE as compared to veratryl alcohol, it can be hypothesized that TvLac5 constitutes the largest fraction and/or most active laccase of the secretome that explains the increased oxidative potential upon adding OPE (Figure 2B). For the remaining extracellular laccase TvLac6 [18], encoded by *LCC6*, only the veratryl alcohol condition showed a significant (*p* < 0.05) transcriptional upregulation, with a log_2_ value of 1.29, whereas the addition of OPE did not result in an altered expression level (Figure 6).

The remaining three laccase genes are indicated to remain intracellular [18] and are thus not expected to contribute to the secreted oxidative potential of *T. versicolor*. While for *LCC4*, only a minor upregulation was observed for both OPE and veratryl alcohol, with log_2_ values of 1.23 and 1.07, respectively, more contrasting effects for OPE and veratryl alcohol were observed for *LCC1* and *LCC7* (Figure 6). Of all the laccase genes, *LCC7* displayed the highest upregulation for the veratryl alcohol condition (log_2_ value of 3.73), while the expression for the OPE condition remained stable (log_2_ value of 0.45). Lastly, a significant transcriptional downregulation was observed for *LCC1* in response to OPE, with a log_2_ value of −2.59, which contrasts with the response to veratryl alcohol, for which a significant upregulation was observed, indicated by a log_2_ value of 2.65.

## 4. Conclusions

In this study, four different *T. versicolor* strains were characterized for their secreted oxidative potential in liquid cultures. While all strains displayed an oxidative potential, strain-specific differences were observed. For *T. versicolor* M99, the addition of 20% v/v of 30 g/L OPE and of 5 mM veratryl alcohol resulted in an increased secreted oxidative potential after 24 h. Furthermore, when cultivated in these inducing conditions for 7 days, the relative gene expression analysis for all individual laccase genes in the strain M99 showed significant transcriptional upregulation of most genes, confirming that the observed oxidative response is laccase-mediated. The highest relative upregulation, namely 300-fold, was observed for *LCC5*, encoding TvLac5, in the presence of OPE. From this observation, it can be concluded that TvLac5 is highly responsive to exogenous inducers, including, but not restricted to, aromatic compounds, and that this laccase contributes most to the observed oxidative potential. It can thus be concluded that OPE is an effective inducer for improving laccase production in *T. versicolor*, thereby potentially leading to a higher efficiency in lignin degradation or to higher yields of extracted laccase when targeting enzyme applications. An additional benefit of OPE is that it does not decrease the colonization rate or biomass production, in contrast to the chemical inducer veratryl alcohol. Given that orange peels, and by extension other citric fruit wastes, constitute a common agro-industrial side stream that is available in high volumes, its use as an inducing agent for laccase production is economically viable.

## Figures and Tables

**Figure 1 jof-10-00370-f001:**
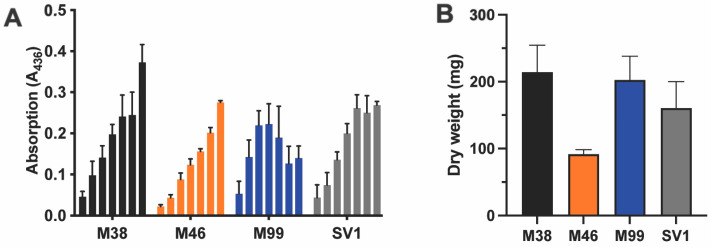
Characterization of the oxidative potential and biomass yield for four different *T. versicolor* strains cultivated in liquid-shaken cultures. (**A**) The oxidative profile, based on ABTS assays performed on samples taken from the culture broth over 7 days, starting 48 h post inoculation, and with each data point representing a measurement taken one day apart. (**B**) Dry weight measurements performed after 9 days of cultivation.

**Figure 2 jof-10-00370-f002:**
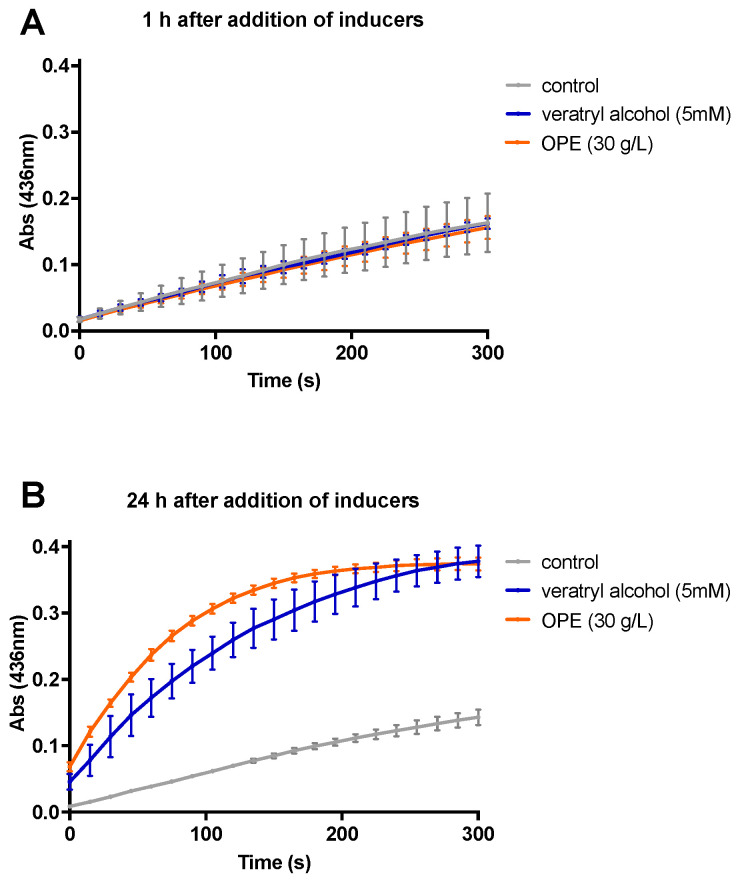
Oxidative potential upon adding 20% OPE or 5 mM veratryl alcohol as an inducer to 7-day old cultures of *T. versicolor* M99. ABTS assay measurements were performed by following the absorption over a time course of 300 s. (**A**) One hour after adding an inducer, (**B**) 24 h after adding an inducer.

**Figure 3 jof-10-00370-f003:**
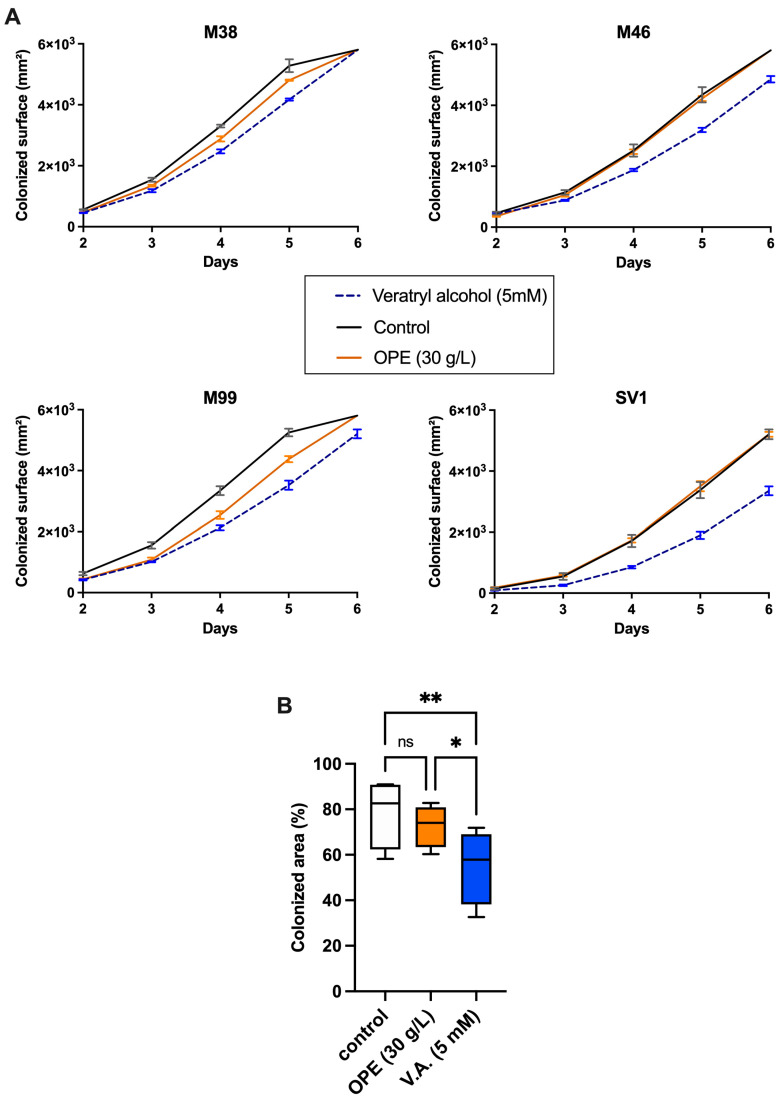
Surface colonization of *T. versicolor* on solid medium. (**A**) Graphical representation of the daily colonized radial area in the presence of OPE (orange) or 5 mM veratryl alcohol (blue) on a solid TDM medium. Error bars represent the standard deviation of 3 biological replicates. (**B**) Boxplot diagram representing the colonized area in % of the total area for all strains on the 5th day of growth. Statistical significance was calculated using a one-way ANOVA analysis, followed by a multiple-comparison Tukey test, with * = *p* ≤ 0.05, ** = *p* ≤ 0.01, ns = not significant.

**Figure 4 jof-10-00370-f004:**
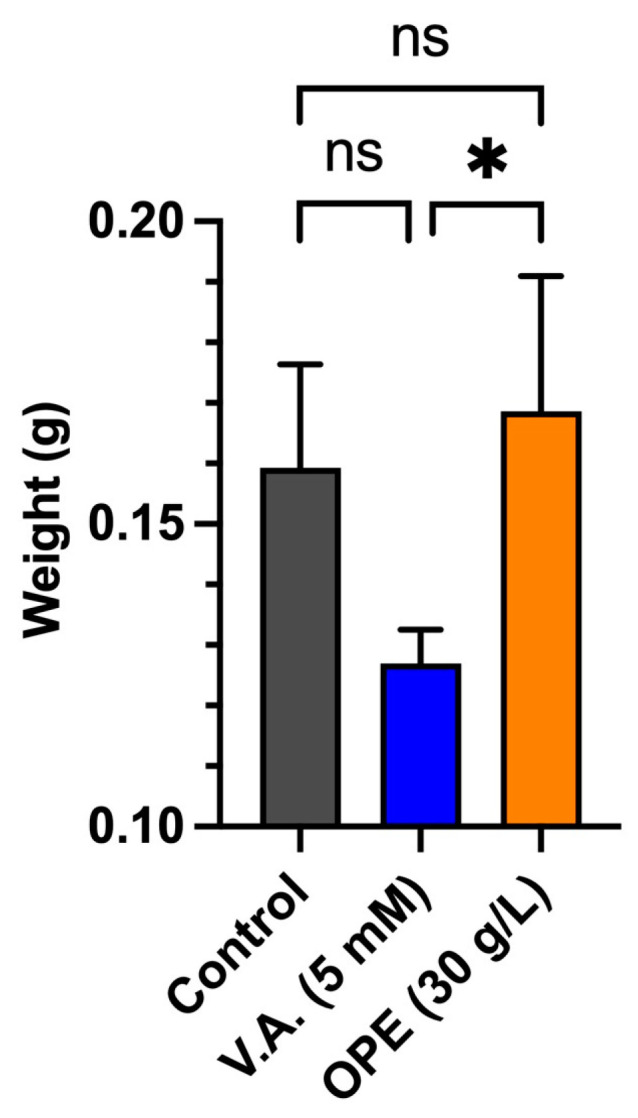
Dry weight of 7-day-old *T. versicolor* cultures cultivated in the presence of laccase inducers. Statistical significance was calculated using a one-way ANOVA analysis, followed by a multiple-comparison Tukey test, with * = *p* ≤ 0.05, ns = not significant.

**Figure 5 jof-10-00370-f005:**
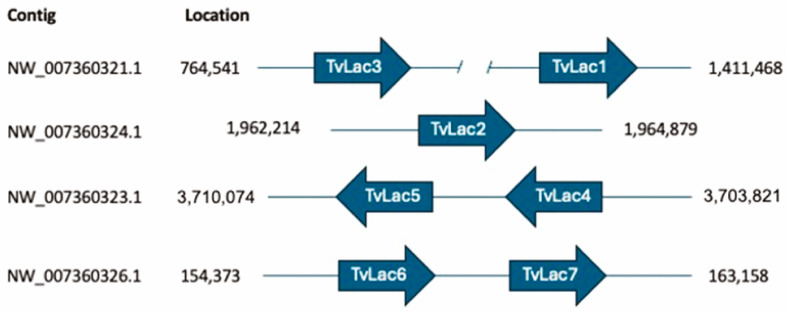
Positioning of laccase genes in the genome of *T. versicolor* FP-101664 SS1.

**Figure 6 jof-10-00370-f006:**
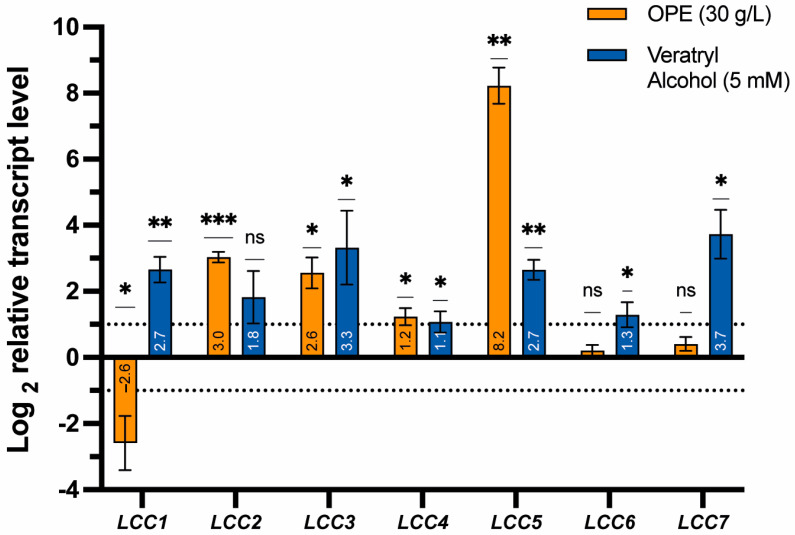
Relative transcription levels of laccase genes (*LCC1*-*LCC7*, representing TvLac1—TvLac7, respectively) from *T. versicolor* M99 cultivated in the presence of 5 mM veratryl alcohol or 30 g/L OPE, versus cultivation in the absence of inducers. Each bar represents the average result of qRT-PCR measurements performed for 3 biological replicates in 2 technical replicates. Error bars represent the standard deviation. The statistical significance of differential expression was assessed using a one sample *t*-test, with * = *p* ≤ 0.05, ** = *p* ≤ 0.01, *** = *p* ≤ 0.001, ns = not significant.

**Table 1 jof-10-00370-t001:** List of different *T. versicolor* strains, with their respective origin and mode of conservation.

Strain	Geographical Origin	Source	Conservation
M38	Czech Republic	Belgian Co-ordinated Collections of Micro-organisms (BCCM)	10% glycerol −130 °C since 12 September 2013
M46	South Korea	Belgian Co-ordinated Collections of Micro-organisms (BCCM)	10% glycerol −130 °C since 16 June 2005
M99	Unknown	Mycelia bvba (reference: *T. versicolor* M9912)	Actively cultured
SV1	Belgium	Isolated from a decaying tree trunk in Schaveys park Linkebeek (Belgium)	Actively cultured

**Table 2 jof-10-00370-t002:** Overview of laccase genes in *T. versicolor* FP-101664 SS1 with sequences of qRT-PCR primers. The housekeeping gene is indicated by *.

Gene Name	Accession Number (NCBI)	Gene Length (bp)	Coding Sequence Length (bp)	Exon #	Sequences of Forward/Reverse qPCR Primers (5′ -> 3′)
*TvLac1*	XM_008034546	2222	1560	11	CATCACGTTGACCGACTGG/GACGTTGATCACAGCAAGCG
*TvLac2*	XM_008038707	2222	1575	11	AAGGCTATCAACTTCGCTTT/CTCATGATTTGCAGCAGAAC
*TvLac3*	XM_008034423	2074	1563	9	ACTTCGGTAACGTCGGGTTC/TGCAGGTTGACCTCGTTGAG
*TvLac4*	XM_008037774	2118	1563	11	CAGATTCTTAGCGGCACCAC/ GAATGTGTGACCGTGCAAGT
*TvLac5*	XM_008037775	2266	1584	12	CCCGTCACCGACTTGACTAT/GGTCTCGTTGGTCAGGTTGT
*TvLac6*	XM_008040042	2266	1539	13	CACCAAGTCGACGGACTTCA/CCCGTAGTGTTAGCACGGTT
*TvLac7*	XM_008040097	2210	1548	13	TTGAACTGTCCATCCCTGGC/GTACCACGGAGAAGGTGTGG
*β-tubulin 2 **	XM_008041613.1	2723	1401	17	ACAGTACCAAGAAGCGACGG/GCCTTCTGGTTGGGATCGAA

## Data Availability

Data are contained within the article and Appendix A.

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
