# Peer review of "Effects of Orange Peel Extract on Laccase Activity and Gene Expression in Trametes versicolor"

_jof, 2024, doi:10.3390/jof10060370_

Round 1

Reviewer 1 Report

The Authors of the Manuscript „ Effects of orange peel extract on the oxidative potential and laccase gene expression in Trametes versicolor” present a study that investigates the effect of crude orange peel extract on the expression of laccases in T. versicolor. The manuscript needs careful rewriting to clarify the motivation and the outcomes. Especially the observation that OPE improves the overall growth of T. versicolor needs to be analyzed more in depth to make the manuscript more substantial. The authors are encouraged to re-submit their manuscript after revision.

Major:1: Is there any rationale except economic reasons why the authors choose to prefer veratryl alcohol as inducing compound for laccases in T. versicolor over coniferyl, sinapyl or paracoumaryl alcohol? Also when looking at Figure 3, surface colonization is reduced when adding veratryl alcohol to such an extent it could also indicate toxicity…

Major 2: There is a lack of linking the compounds found in orange peels (line 285- 289) to the inductive effect described here. This is clearly a knowledge gap and could also be addressed in a follow-up study by the authors. However, the authors are encouraged to narrow down the compounds listed, if possible.

Major 3: It is understandable that a defined substance or an underused side stream is preferred for an application scenario, however it might have been good to quantitatively evaluate the expression of laccases on T. versicolor also by testing a “classical” laccase substrate such as soluble lignin sulfonate. Can the authors please comment why they did not consider this for their study?

Major 4: Line 270 - The claim that laccases are clustered in pairs is not evident from the content presented in Table 2. -The authors should either indicate gene positions on the contigs or place the reference to the table at a different location within the sentence to avoid confusion.

Major 5: Figure 4 – The observed growth differences should also be determined gravimetrically and related to the consumed total carbon present in the different media used. Especially when the authors have used cell dry weight determination as shown in figure 1, it should be possible to apply the same method for different media. This should be included in the manuscript. If no analytics of total carbon is possible, at least the (partial) consumption of the 10 g/L glucose of the TDM should be monitored throughout the cultivation.

Major 6: Section 2.5 (Bioinformatics analysis) – The Multiple sequence alignment can be removed from the main body of the manuscript and integrated into the text of the Supplementary Figure A2 since this is a pretty basic technique used and does not add much to the story except verifying that all laccase genes have retained the ability to bind copper.

Major 7: Why is there no time-course data shown for the effect of OPE or V.A. on the oxidative potential over the cultivation period for time points later than 6 days? Also, it appears that the authors performed 2 types of experiments to investigate the effect of OPE and V.A. A) Cultivation in media with the respective formulation for the entire cultivation time and B) addition of OPE and V.A. after a preculture phase and measuring the inductive effect (shown in figure 2). This has to be clarified in the manuscript and in the figure descriptions.

Major 8: Line 196: Figure 1A does show more than 2 sample time points per strain, hence more samples than day 3 and 9 have been analyzed. Please rewrite this sentence to clarify and align with figure 1.

Minor:

Line 31: It might be better for the non-expert reader to explain briefly where the dubbing “blue” multi-copper oxidase originates from and that it refers to the structure of the copper centre of the enzyme.

Line 97: Please double-check the use of the term peeling and peelings in the context of the fruit peel. It might be more appropriate to simply use orange peel or orange peels.

.

Author Response

We gratefully acknowledge the reviewer for the valuable input, which helped us to improve the manuscript. Below you can find our point-to-point responses indicated. Line numbers refer to the revised version of the manuscript.

Major comments

Major 1: Is there any rationale except economic reasons why the authors choose to prefer veratryl alcohol as inducing compound for laccases in T. versicolor over coniferyl, sinapyl or paracoumaryl alcohol? Also when looking at Figure 3, surface colonization is reduced when adding veratryl alcohol to such an extent it could also indicate toxicity…

Author response: We have selected veratryl alcohol as an inducing compound as a reference for the study of the effects of orange peels as it has been studied extensively in relation to laccases activity in white rot fungi (WRF). This selection is supported by several studies in which a link was reported between veratryl alcohol and the overexpression and increased production of laccases (e.g. reference 25 in the paper). Indeed, there are many other compounds that induce laccase expression in WRF, as indicated by the reviewer, but based on the described effects each one of these inducing compounds could find use in our study as a reference. Only one compound was needed, given that this is a reference and the focus is placed on orange peels and there was no rationale for choosing veratryl alcohol. In the revised version we have rewritten the text so that it is more clear that the investigation is focused on orange peel extract (OPE) and that veratryl alcohol is a reference, as follows: “Subsequently, we investigated the effect of the addition of an OPE preparation on the oxidative potential of T. versicolor (Figure 2). … As a reference, 5 mM veratryl alcohol was tested as well, which is a well-known inducer of laccase expression in WRF [25, 36]." (lines 218-224).  We agree with the reviewer that the chosen veratryl alcohol concentration causes growth inhibition, as also explained in the manuscript text as follows: “In other WRF species, veratryl alcohol or its oxidized form veratrylaldehyde causes a growth-inhibiting effect at concentrations of 5 mM or higher [25, 26]. To investigate this for T. versicolor and to compare this to OPE, we analyzed the effect of either 5 mM veratryl alcohol or 20% OPE on growth on solid medium by measuring the radial colonization rate (Figure 3).” (lines 242-246), “Upon the addition of veratryl alcohol, a systematic decrease in colonization rate was observed for all four strains (Figure 3A).” (lines 247-248), “In the presence of 5 mM veratryl alcohol, a significant decrease (p<0.01) in colonization rate was observed as compared to the control condition (Figure 3B).” (lines 252-254) and “After 7 days of cultivation, cell dry weights were determined for the biomass produced in the different liquid cultures (Figure 4). This demonstrated a significant decrease (p ≤ 0.05) in biomass for cultures in the presence of 5 mM veratryl alcohol as compared to cultures in the presence of OPE.” (lines 259-262). Nevertheless, with the colonized area reaching almost 60% on average, there is still sufficient biomass formation so that it is not counterproductive in its role to induce overall laccase production. This corresponds to a previous study, in which the same concentration was selected (5 mM) and reported to have the highest induction effect without impacting biomass too much (Dekker et al (2002) Enzyme and Microbial Technology 30(3), 374-380).

Major 2: There is a lack of linking the compounds found in orange peels (line 285- 289) to the inductive effect described here. This is clearly a knowledge gap and could also be addressed in a follow-up study by the authors. However, the authors are encouraged to narrow down the compounds listed, if possible.

Author response: We have added more information on the composition of orange peels and its water-extractable compounds, based on existing literature that describes detailed analyses of orange peel compositions. In addition, we have extended the discussion on hypothesizing which compounds in OPE might be responsible for the observed laccase induction, including the likelihood of limonene influencing the regulation of laccase expression. This information is added in sections 3.3. and 3.4. as follows: “This growth-supporting role is enabled through the extraction of various reduced sugars and nutritive components during the hot water treatment of the OPE preparation. Orange peels are mainly composed of cellulose, which breaks down into glucose and fructose when hydrolyzed [46]. According to Yeoh et al., hot water treatment of orange peels enables to yield around 5% pectin of the dry basis, which is also a source of reduced sugars when hydrolyzed [47].” (lines 270-275) and “orange peels are rich in aromatic compounds, including free phenolic compounds from two main classes: flavonones and flavones. Moreover, large amounts of phytochemicals such as carotenoids, flavonoids, aldehydes, esters, terpenes, alcohols and ketones are also present in orange peels [48, 49]. For example, limonene constitues around 75% of the total extractable essential oils [46] and is therefore likely to be involved in influencing laccase expression, as previously observed by Böhmer et al. [50]. This can be explained by the structural resemblance between limonene and compounds found in lignin. Given the complex composition of OPE, it is difficult to exactly quantify the precise involvement of each of each of its components in the more pronounced transcriptional induction.” (lines 322-330). We agree with the reviewer that a deeper analysis on the individual components found in OPE and an identification of those that influence laccase regulation would present an interesting topic for a follow-up study.

Major 3: It is understandable that a defined substance or an underused side stream is preferred for an application scenario, however it might have been good to quantitatively evaluate the expression of laccases on T. versicolor also by testing a “classical” laccase substrate such as soluble lignin sulfonate. Can the authors please comment why they did not consider this for their study?

Author response: We agree that it might have been an interesting avenue to quantitively evaluate other laccase-inducing compounds such as lignin sulfonate. However, the goal of this study was to investigate a more economically practical avenue by inducing laccase production using an industrial waste stream, as also rightfully noted by the reviewer. In the context of the aim of evaluating OPE as a potential industrial side stream that can be upcycled for the production of laccases and fungal biomass, we focused on OPE and performed a quantitative analysis of its inducing effects on the expression of the 7 laccase genes present in the T. versicolor genome. The sole reason for including an additional inducing compound in the study was to have a well-described reference. Veratryl alcohol was chosen because it is a pure compound with low molecular weight that can be used in experiments across different studies in a homogenous preparation and leading to reproducible results. In contrast, lignin sulfonate is characterized by a very broad molecular mass range (1,000 to 140,000 Da) and although it would be interesting to study its effect on laccase expression, this is beyond the scope of this paper and its use as a reference is less suitable as compared to veratryl alcohol.

Major 4: Line 270 - The claim that laccases are clustered in pairs is not evident from the content presented in Table 2. -The authors should either indicate gene positions on the contigs or place the reference to the table at a different location within the sentence to avoid confusion.

Author response: We thank the reviewer for this suggestion and have added a graphical schematic overview of the synteny of genes encoding laccase isozymes in the genome of T. versicolor, with indications of gene locations, as a new figure in the revised version of the manuscript (Figure 5).

Major 5: Figure 4 – The observed growth differences should also be determined gravimetrically and related to the consumed total carbon present in the different media used. Especially when the authors have used cell dry weight determination as shown in figure 1, it should be possible to apply the same method for different media. This should be included in the manuscript. If no analytics of total carbon is possible, at least the (partial) consumption of the 10 g/L glucose of the TDM should be monitored throughout the cultivation.

Author response: We agree with the comment that biomass should be represented quantitively and we have therefore repeated this experiment and measured cell dry weight gravimetrically. The results are now presented in a revised Figure 4, while the previous qualitative image has been removed from the manuscript. The reason for this is that we used a more accurate inoculum preparation approach to prepare the liquid cultures, starting with blended mycelium instead of agar slices and that the newly obtained quantified results were not entirely in agreement with the previously presented visual assessment of mycelial pellet formation, but more in agreement with the colonization rate experiments on solid media. Given the low reliability of the previous result, we decided to remove it from the manuscript. The manuscript text has been adapted accordingly, as follows: “After 7 days of cultivation, cell dry weights were determined for the biomass produced in the different liquid cultures (Figure 4). This demonstrated a significant decrease (p ≤ 0.05) in biomass for cultures in the presence of 5 mM veratryl alcohol as compared to cultures in the presence of OPE. Also, the average mycelium dry weight appeared higher for the control condition as compared to the condition with veratryl alcohol but without being supported by statistical significance." (lines 259-264).  We respectfully disagree with the suggestion to monitor glucose depletion in the medium over time and did not take up on this suggestion as OPE contains many different reduced sugars in addition to the 10 g/L glucose in the TDM medium, leading to an undefined composition and possible difficulties in interpretation. Performing glucose measurements between the veratryl alcohol and control condition would not contribute much to the study, as veratryl alcohol is used as a reference for OPE.  

Major 6: Section 2.5 (Bioinformatics analysis) – The Multiple sequence alignment can be removed from the main body of the manuscript and integrated into the text of the Supplementary Figure A2 since this is a pretty basic technique used and does not add much to the story except verifying that all laccase genes have retained the ability to bind copper.

Author response: We agree with the reviewer’s comment and have moved the multiple sequence alignment to Supplementary Materials (now Supplementary Figure S3).

Major 7: Why is there no time-course data shown for the effect of OPE or V.A. on the oxidative potential over the cultivation period for time points later than 6 days? Also, it appears that the authors performed 2 types of experiments to investigate the effect of OPE and V.A. A) Cultivation in media with the respective formulation for the entire cultivation time and B) addition of OPE and V.A. after a preculture phase and measuring the inductive effect (shown in figure 2). This has to be clarified in the manuscript and in the figure descriptions.

Author response: In the context of an industrial application, cultivation time is an economically sensitive parameter and we therefore concluded that a growth period of 7 days generated sufficient biomass to investigate the laccase inducing effect of OPE and veratryl alcohol. We have adapted the text in the revised manuscript, both in the main text as in the figure legends, thereby clarifying the differences between the two different experimental set-ups, as follows:  “A daily measurement was performed, starting on day 3 post inoculation and lasting until day 9 (Figure 1A).” (lines 207-208), “Subsequently, we investigated the effect of the addition of an OPE preparation on the oxidative potential of T. versicolor (Figure 2). … The inducers were added to 7-day old shaken cultures of the strain. In contrast to the previous set-up, the oxidative potential was measured over a shorter time frame, after 1 and 24 hours after the addition of the inducer, and kinetic measurements were made (Figure 2)." (lines 218-227), “Figure 2. Oxidative potential upon adding 20% OPE or 5 mM veratryl alcohol as an inducer to 7-day old cultures of T. versicolor M99. ABTS assay measurements were performed by following the absorption over a time course of 300 seconds. A. 1 hour after adding an inducer, B. 24 hours after adding an inducer.” (lines 230-233).

Major 8: Line 196: Figure 1A does show more than 2 sample time points per strain, hence more samples than day 3 and 9 have been analyzed. Please rewrite this sentence to clarify and align with figure 1.

Author response: We have adapted this sentence to make it more comprehensible as follows: “A daily measurement was performed, starting on day 3 post inoculation and lasting until day 9 (Figure 1A).” (lines 207-208).

Minor:

Line 31: It might be better for the non-expert reader to explain briefly where the dubbing “blue” multi-copper oxidase originates from and that it refers to the structure of the copper centre of the enzyme.

Author response: We have included an informative sentence on the origin of the term “blue” that characterises laccases, as follows: “Laccases form a family of blue multi-copper oxidases, named after the intense blue colour associated with the notable absorption spectrum arising from their multi-copper core, which underlies the oxidative potential [4].” (lines 33-36).

Line 97: Please double-check the use of the term peeling and peelings in the context of the fruit peel. It might be more appropriate to simply use orange peel or orange peels.

Author response: We have changed the term “peelings” to “peels” throughout the manuscript.

Reviewer 2 Report

In this article, the author explores and attempts to use orange peeling extract (OPE) as an inducer for Trametes versicolor, finding that it can promote growth and has a particularly significant effect on the expression of certain laccases. This study represents an intriguing experiment with positive implications for developing the potential uses of agricultural waste and new strategies for high-yield laccase production. However, the paper lacks analysis of the components of OPE and the main effective components that promote laccase. Further refinement by the author is recommended. Overall, major revision is inevitable, and some suggestions are provided below for consideration.

1.      For White rot fungi (WRF), after its first mention, the author could consider using the abbreviation WRF for subsequent references. This approach is more concise and clear, and it also aligns with common practice.

2.      Introduction, Line 26-30. This section requires further improvement, particularly with additional background information on white-rot fungi (WRF). Below is my attempt at revising these for clarity, provided for your reference (The recommended citations are also listed).

Trametes versicolor is a species of fungus that belongs to the Polyporaceae family and is a member of the white rot fungi (WRF) group. WRF are celebrated for their natural decomposing abilities, capable of breaking down both lignin and cellulose, the major biopolymers found in lignocellulosic biomass [Ecotoxicology and Environmental Safety, 254(2023) 114697]. These fungi are distinguished by their oxidative and extracellular ligninolytic systems, which are characterized by low substrate specificity. This property enables them to address and break down various environmental contaminants [Chemosphere, 301(2022), 134776; Journal of cleaner production, 354(2022), 131681]. The ligninolytic systems of WRF comprise several key extracellular enzymes, namely dye-decolorizing peroxidase (EC 1.11.1.19, DyP), lignin peroxidase (EC 1.11.1.14, LiP), manganese peroxidase (EC 1.11.1.13, MnP), versatile peroxidase (EC 1.11.1.16, VP), and laccase (EC 1.10.3.2, Lac) [Bioresource Technology, 395 (2024), 130337]. Laccases form a family…”

3.      In lines 67-78, the author should provide a comprehensive introduction to the laccase gene isoforms of T. versicolor, including the number of laccase isoforms, and appropriately compare this with other typical white rot fungi. A review article has already conducted a systematic summary on this issue, which the author could refer to and pay attention to: Science of The Total Environment, 778 (2021), 146132.

4.      On line 93, please display tables in the text using the three-line table format. The same applies to other Tables.

5.      Lines 96-98 contain ambiguity and require modification.

6.      In Section 2.6, "Quantitative Reverse Transcriptase PCR," it is recommended that the author includes primer information in the main text rather than in supplementary materials. Additionally, it is suggested to supplement the supplementary materials with gel electrophoresis images of qualitative amplification using qRT-PCR primers. This can assist in assessing the amplification effectiveness and specificity.

7.      In Section 3.1, "Growth and oxidative potential of four different T. versicolor strains in liquid cultures," ABTS is typically used as a specific substrate to determine laccase activity. However, the focus in the article is on oxidative potential rather than laccase activity. Is this due to consideration of the oxidizing effects of other possible oxidases on ABTS? It is recommended that the author discuss and explain this in the text.

8.      On line 192, the text "T. versicolor" should be italicized, and the same formatting should be applied to other occurrences.

9.      For Figure 2, although the author has explained the meaning of each curve in the caption "(grey = control, orange = OPE, blue = 5 mM veratryl alcohol)," it is still recommended to annotate the different colored curves with labels in the top right corner of the image for clarity. It is also suggested to make similar adjustments to other figures.

10.   The markers indicating significant differences are missing in Figure 3B.

11.   In Figure 4, where the author has used images of liquid culture in conical flasks to show the effects of different inducers on fungal growth, it is recommended to move this figure to the supplementary materials section. Instead, use a graph showing the change in fungal biomass dry weight over time. This approach would be more rigorous and conducive to quantitative analysis.

12.   In Figure 7, the OPE treatment groups for LCC6 and LCC7 show significant differences compared to the control. Why are they marked as having no significant difference?

13.   In this paper, orange peeling extract (OPE) has demonstrated its capability to support the growth of T. versicolor and act as a laccase inducer. What specific components does OPE contain, and what are the primary components that contribute to the growth support and laccase induction in T. versicolor? It is suggested that the author analyze the components of OPE or engage in a deeper discussion on this topic.

14.   There are several spelling, grammatical errors, issues with phrasing, and formatting errors throughout the document. It is recommended that the author carefully read through the entire manuscript to make corrections, preventing similar errors. Additionally, it is suggested to consider engaging a professional language service company to revise and proofread the English in the article.

Author Response

We gratefully acknowledge the reviewer for the valuable input, which helped us to improve the manuscript. Below you can find our point-to-point responses indicated. Line numbers refer to the revised version of the manuscript.

Major comments

  1. For White rot fungi (WRF), after its first mention, the author could consider using the abbreviation WRF for subsequent references. This approach is more concise and clear, and it also aligns with common practice.

Author response: We have implemented this throughout the manuscript.

  1. Introduction, Line 26-30. This section requires further improvement, particularly with additional background information on white-rot fungi (WRF). Below is my attempt at revising these for clarity, provided for your reference (The recommended citations are also listed).

Trametes versicolor is a species of fungus that belongs to the Polyporaceae family and is a member of the white rot fungi (WRF) group. WRF are celebrated for their natural decomposing abilities, capable of breaking down both lignin and cellulose, the major biopolymers found in lignocellulosic biomass [Ecotoxicology and Environmental Safety, 254(2023) 114697]. These fungi are distinguished by their oxidative and extracellular ligninolytic systems, which are characterized by low substrate specificity. This property enables them to address and break down various environmental contaminants [Chemosphere, 301(2022), 134776; Journal of cleaner production, 354(2022), 131681]. The ligninolytic systems of WRF comprise several key extracellular enzymes, namely dye-decolorizing peroxidase (EC 1.11.1.19, DyP), lignin peroxidase (EC 1.11.1.14, LiP), manganese peroxidase (EC 1.11.1.13, MnP), versatile peroxidase (EC 1.11.1.16, VP), and laccase (EC 1.10.3.2, Lac) [Bioresource Technology, 395 (2024), 130337]. Laccases form a family…”

Author response: We thank the reviewer for the suggestion. In response, we have extended the Introduction with more elaborate background descriptions on WRF, thereby implementing some of the suggested modifications, as follows: “The ligninolytic systems of WRF comprise several key extracellular enzymes, namely dye-decolorizing peroxidase (EC 1.11.1.19, DyP), lignin peroxidase (EC 1.11.1.14, LiP), manganese peroxidase (EC 1.11.1.13, MnP), versatile peroxidase (EC 1.11.1.16, VP), and laccase (EC 1.10.3.2, Lac) [3]. Laccases form a family of blue multi-copper oxidases named after the intense blue colour associated with the notable absorption spectrum arising from their multi-copper core, which underlies the oxidative potential [4].” (lines 30-36). We have added the following references as suggested by the reviewer: Chen, S, Zhu, M, Guo, X, Yang, B., Zhuo, R. Coupling of Fenton reaction and white rot fungi for the degradation of organic pollutants. Ecotoxicology and Environmental Safety 2023, 254, 114697. 10.1016/j.ecoenv.2023.114697 (reference 1) and Gao, X, Wei, M, Zhang, X, Xun, Y, Duan, M, Yang, Z, Zhu, M, Zhu, Y, Zhuo, R. Copper removal from aqueous solutions by white rot fungus Pleurotus ostreatus GEMB-PO1 and its potential in co-remediation of copper and organic pollutants. Biores Techn. 2024, 395, 130337 (reference 3).

  1. In lines 67-78, the author should provide a comprehensive introduction to the laccase gene isoforms of T. versicolor, including the number of laccase isoforms, and appropriately compare this with other typical white rot fungi. A review article has already conducted a systematic summary on this issue, which the author could refer to and pay attention to: Science of The Total Environment, 778 (2021), 146132.

Author response: We acknowledge the reviewer’s comment to provide additional information on the laccase isoforms. However, in our opinion a lack of consistent annotation of laccase isoforms in literature makes it difficult to perform a comparative analysis of the isoforms with respect to other WRF species, as it might lead to speculations/errors when such an analysis is not properly done, which is furthermore beyond the scope of our work. Nevertheless, we have now mentioned the number of laccase isoforms in T. versicolor in the Introduction, as requested by the reviewer, as follows: “With 7 laccase-encoding genes, T. versicolor has been recognized as one of the most effective WRF for large-scale laccase production [28, 29].” (lines 72-73). In addition, we have also added the suggested reference (reference 6).

  1. On line 93, please display tables in the text using the three-line table format. The same applies to other Tables.

Author response: We have implemented the reviewer’s remark for all tables.

  1. Lines 96-98 contain ambiguity and require modification.

Author response: We have rephrased this segment to remove ambiguity and improve clarity, as follows: “When indicated, one of the following supplements were added: 20 % OPE or 5 mM veratryl alcohol (Sigma-Aldrich). OPE was prepared by autoclaving 30 g wet orange peels in 1 L double-distilled water (ddH2O) for 15 minutes.” (lines 103-105).

  1. In Section 2.6, "Quantitative Reverse Transcriptase PCR," it is recommended that the author includes primer information in the main text rather than in supplementary materials. Additionally, it is suggested to supplement the supplementary materials with gel electrophoresis images of qualitative amplification using qRT-PCR primers. This can assist in assessing the amplification effectiveness and specificity.

Author response: We have implemented the reviewer’s request and moved the primer information from the Supplementary Materials to Table 2. An image of the gel electrophoresis analysis of qPCR products has been added to the Supplementary Materials (Supplementary Figure S2).

  1. In Section 3.1, "Growth and oxidative potential of four different T. versicolor strains in liquid cultures," ABTS is typically used as a specific substrate to determine laccase activity. However, the focus in the article is on oxidative potential rather than laccase activity. Is this due to consideration of the oxidizing effects of other possible oxidases on ABTS? It is recommended that the author discuss and explain this in the text.

Author response: We acknowledge the reviewer’s comment and have rephrased sentences in the manuscript in which oxidative potential was mentioned when referring to specific laccase activity. Also, we have now explicitly written in the manuscript that the observed oxidation potential is at least in part due to laccase activity, given the results of our transcriptional analysis, as follows: “when cultivated in these inducing conditions for 7 days, relative gene expression analysis for all individual laccase genes in strain M99 showed significant transcriptional upregulation of most genes, confirming that the observed oxidative response is laccase-mediated.” (lines 378-381).

  1. On line 192, the text "T. versicolor" should be italicized, and the same formatting should be applied to other occurrences.

Author response: This has been adjusted.

  1. For Figure 2, although the author has explained the meaning of each curve in the caption "(grey = control, orange = OPE, blue = 5 mM veratryl alcohol)," it is still recommended to annotate the different colored curves with labels in the top right corner of the image for clarity. It is also suggested to make similar adjustments to other figures.

Author response: Symbol legends have been added to the graphs themselves in Figures 2 and 3. Such a symbol legend was already present in Figure 6 (previously Figure 5).

  1. The markers indicating significant differences are missing in Figure 3B.

Author response: Standard deviations are shown on the graph in Figure 3B.

  1. In Figure 4, where the author has used images of liquid culture in conical flasks to show the effects of different inducers on fungal growth, it is recommended to move this figure to the supplementary materials section. Instead, use a graph showing the change in fungal biomass dry weight over time. This approach would be more rigorous and conducive to quantitative analysis.

Author response: We agree with the comment that biomass should be represented quantitively and we have therefore repeated this experiment and measured cell dry weight gravimetrically. The results are now presented in a revised Figure 4, while the previous qualitative image has been removed from the manuscript. The reason for this is that we used a more accurate inoculum preparation approach to prepare the liquid cultures, starting with blended mycelium instead of agar slices and that the newly obtained quantified results were not entirely in agreement with the previously presented visual assessment of mycelial pellet formation, but more in agreement with the colonization rate experiments on solid media. Given the low reliability of the previous result, we decided to remove it from the manuscript. The manuscript text has been adapted accordingly, as follows: “After 7 days of cultivation, cell dry weights were determined for the biomass produced in the different liquid cultures (Figure 4). This demonstrated a significant decrease (p ≤ 0.05) in biomass for cultures in the presence of 5 mM veratryl alcohol as compared to cultures in the presence of OPE. Also, the average mycelium dry weight appeared higher for the control condition as compared to the condition with veratryl alcohol but without being supported by statistical significance." (lines 259-264). 

  1. In Figure 7, the OPE treatment groups for LCC6 and LCC7 show significant differences compared to the control. Why are they marked as having no significant difference?

Author response: We believe the reviewer might has misinterpreted the graph: the OPE treatment groups (orange bars) did not display any significant difference for LCC6 and LCC7, while the veratryl alcohol treatment groups (blue bars) did. it is standard practice for gene regulation data to use the control condition as the baseline and thus not represent it directly in the bar chart. We checked our data again and confirm that for the OPE condition and confirm that there is no significant differential expression as compared to the control condition for LCC6 and LCC7.

  1. In this paper, orange peeling extract (OPE) has demonstrated its capability to support the growth of T. versicolor and act as a laccase inducer. What specific components does OPE contain, and what are the primary components that contribute to the growth support and laccase induction in T. versicolor? It is suggested that the author analyze the components of OPE or engage in a deeper discussion on this topic.

Author response: We have added more information on the composition of orange peels and its water-extractable compounds, based on existing literature that describes detailed analyses of orange peel compositions. In addition, we have extended the discussion on hypothesizing which compounds in OPE might be responsible for the observed laccase induction, including the likelihood of limonene influencing the regulation of laccase expression. This information is added in sections 3.3. and 3.4. as follows: “This growth-supporting role is enabled through the extraction of various reduced sugars and nutritive components during the hot water treatment of the OPE preparation. Orange peels are mainly composed of cellulose, which breaks down into glucose and fructose when hydrolyzed [46]. According to Yeoh et al., hot water treatment of orange peels enables to yield around 5% pectin of the dry basis, which is also a source of reduced sugars when hydrolyzed [47].” (lines 270-275) and “orange peels are rich in aromatic compounds, including free phenolic compounds from two main classes: flavonones and flavones. Moreover, large amounts of phytochemicals such as carotenoids, flavonoids, aldehydes, esters, terpenes, alcohols and ketones are also present in orange peels [48, 49]. For example, limonene constitues around 75% of the total extractable essential oils [46] and is therefore likely to be involved in influencing laccase expression, as previously observed by Böhmer et al. [50]. This can be explained by the structural resemblance between limonene and compounds found in lignin. Given the complex composition of OPE, it is difficult to exactly quantify the precise involvement of each of each of its components in the more pronounced transcriptional induction.” (lines 322-330).

  1. There are several spelling, grammatical errors, issues with phrasing, and formatting errors throughout the document. It is recommended that the author carefully read through the entire manuscript to make corrections, preventing similar errors. Additionally, it is suggested to consider engaging a professional language service company to revise and proofread the English in the article.

Author response: The manuscript has been carefully proofread by different people and we have corrected all spelling and grammatical errors.

Reviewer 3 Report

Laccases form a family of blue multi-copper oxidases and have a large potential commercial value for their potential of oxidizing a wide range of recalcitrant compounds. Trameter versicolor has been a well-known laccase producing fungal species as a typical white-rotting fungi. This research aimed to increase laccase production adding grounded orange peelings as inducing agent and has a practical significance. However, there is some errors as following:

1.It is incomplete that only M99 strain has bee selected in the experiment about addition of OPE preparation or 5 mM veratryl alcohol. Although It showed slightly difference in the result of characterization of oxidative potential, for showing its effect as inducing agent of producing laccase, it will be more reliable that 4 different origin strains are all involved the experiment of the addition of orange peel extract and veratryl alcohol Impacting on the oxidative potential.  

2.It is low scientific that the result was showed with picture of the number of pellets not hyphae biomass of T. versicolors growth in liquid medium by cultivating under three different condition.  And there is a lacking of the liquid cultivating experiment with only OPE. In addition, the same question, why only M99 strain?

3.There is a regret that authors did not provide the component measuring of experimental OPE and could not investigate the exact component which affect the oxidative potential of T. versicolor

1. In figure 2, the represent meaning of different colour line should be marked clearly in the figure, not in the legend.

2. In figure 3, the meaning of "ns" showed be illustrated in the legend.

Author Response

We gratefully acknowledge the reviewer for the valuable input, which helped us to improve the manuscript. Below you can find our point-to-point responses indicated. Line numbers refer to the revised version of the manuscript.

Major comments

1.It is incomplete that only M99 strain has been selected in the experiment about addition of OPE preparation or 5 mM veratryl alcohol. Although It showed slightly difference in the result of characterization of oxidative potential, for showing its effect as inducing agent of producing laccase, it will be more reliable that 4 different origin strains are all involved the experiment of the addition of orange peel extract and veratryl alcohol Impacting on the oxidative potential. 

Author response: In the revised version of the manuscript we have added a more clear explanation of why strain M99 was selected for the investigation of inducer effects: “This experiment was performed for strain M99 of T. versicolor, which was selected because of its decreasing oxidative activity after an incubation of 7 days and longer (Figure 1A), thus representing an interesting candidate for further optimization by the induction of laccase production.” (lines 219-222). Given that the main objective of the work was to focus on a more in-depth characterization of the induction of laccase expression and its contribution to the oxidative potential in response to the addition of orange peel extract, instead of investigating inter-strain differences, we don’t fully agree with the comment of the reviewer that the induction experiments should be performed for all 4 strains.

2.It is low scientific that the result was showed with picture of the number of pellets not hyphae biomass of T. versicolors growth in liquid medium by cultivating under three different condition.  And there is a lacking of the liquid cultivating experiment with only OPE. In addition, the same question, why only M99 strain?

Author response: We agree with the comment that biomass should be represented quantitively and we have therefore repeated this experiment and measured cell dry weight gravimetrically. The results are now presented in a revised Figure 4, while the previous qualitative image has been removed from the manuscript. The reason for this is that we used a more accurate inoculum preparation approach to prepare the liquid cultures, starting with blended mycelium instead of agar slices and that the newly obtained quantified results were not entirely in agreement with the previously presented visual assessment of mycelial pellet formation, but more in agreement with the colonization rate experiments on solid media. Given the low reliability of the previous result, as rightfully mentioned by the reviewer, we decided to remove it from the manuscript. The manuscript text has been adapted accordingly, as follows: “After 7 days of cultivation, cell dry weights were determined for the biomass produced in the different liquid cultures (Figure 4). This demonstrated a significant decrease (p ≤ 0.05) in biomass for cultures in the presence of 5 mM veratryl alcohol as compared to cultures in the presence of OPE. Also, the average mycelium dry weight appeared higher for the control condition as compared to the condition with veratryl alcohol but without being supported by statistical significance." (lines 259-264).  We have added an explanation to the manuscript why strain M99 was selected for the investigation of inducer effects: “This experiment was performed for strain M99 of T. versicolor, which was selected because of its decreasing oxidative activity after an incubation of 7 days and longer (Figure 1A), thus representing an interesting candidate for further optimization by the induction of laccase production.” (lines 219-222). Given that the main objective of the work was to focus on a more in-depth characterization of the induction of laccase expression and its contribution to the oxidative potential in response to the addition of orange peel extract, instead of investigating inter-strain differences, we don’t fully agree with the comment of the reviewer that the induction experiments should be performed for all 4 strains.

3.There is a regret that authors did not provide the component measuring of experimental OPE and could not investigate the exact component which affect the oxidative potential of T. versicolor.

Author response: We have added more information on the composition of orange peels and its water-extractable compounds, based on existing literature that describes detailed analyses of orange peel compositions. In addition, we have extended the discussion on hypothesizing which compounds in OPE might be responsible for the observed laccase induction, including the likelihood of limonene influencing the regulation of laccase expression. This information is added in sections 3.3. and 3.4. as follows: “This growth-supporting role is enabled through the extraction of various reduced sugars and nutritive components during the hot water treatment of the OPE preparation. Orange peels are mainly composed of cellulose, which breaks down into glucose and fructose when hydrolyzed [46]. According to Yeoh et al., hot water treatment of orange peels enables to yield around 5% pectin of the dry basis, which is also a source of reduced sugars when hydrolyzed [47].” (lines 270-275) and “orange peels are rich in aromatic compounds, including free phenolic compounds from two main classes: flavonones and flavones. Moreover, large amounts of phytochemicals such as carotenoids, flavonoids, aldehydes, esters, terpenes, alcohols and ketones are also present in orange peels [48, 49]. For example, limonene constitues around 75% of the total extractable essential oils [46] and is therefore likely to be involved in influencing laccase expression, as previously observed by Böhmer et al. [50]. This can be explained by the structural resemblance between limonene and compounds found in lignin. Given the complex composition of OPE, it is difficult to exactly quantify the precise involvement of each of each of its components in the more pronounced transcriptional induction.” (lines 322-330).

  1. In figure 2, the represent meaning of different colour line should be marked clearly in the figure, not in the legend.

Author response: We have implemented this suggestion: symbol legends have been added to the graphs themselves in Figures 2 and 3.

  1. In figure 3, the meaning of “ns” showed be illustrated in the legend.

Author response: We have implemented this suggestion and added the meaning of “ns” to the legends of Figures 3, 4 and 6.

Round 2

Reviewer 1 Report

The Authors have made convincing improvements on the points raised during the review - I hereby reccomend to accept the manuscript in the existing form of it as submitted now.

The Authors have made convincing improvements on the points raised during the review - I hereby reccomend to accept the manuscript in the existing form of it as submitted now.

Reviewer 2 Report

As the reviewer, I acknowledge and appreciate the author's response to my concerns, as well as the subsequent revisions made to the manuscript. The changes address the issues raised effectively, and I believe that the manuscript now meets the publication standards. Therefore, I agree to accept the article for publication.

No further suggestions.

Reviewer 3 Report

Laccases form a family of blue multi-copper oxidases and have a large potential commercial value for their potential of oxidizing a wide range of recalcitrant compounds. Trameter versicolor has been a well-known laccase producing fungal species as a typical white-rotting fungi. This research aimed to increase laccase production adding grounded orange peelings as inducing agent and has a practical significance. According to the questions, authors provided reply in details and the results seems more scientific and reliable.

Now the tables and figures are more reasonable.